# Moving the Lab into the Mountains: A Pilot Study of Human Activity Recognition in Unstructured Environments

**DOI:** 10.3390/s21020654

**Published:** 2021-01-19

**Authors:** Brian Russell, Andrew McDaid, William Toscano, Patria Hume

**Affiliations:** 1Sports Performance Institute, Auckland University of Technology, Auckland 0632, New Zealand; patria.hume@aut.ac.nz; 2Department of Mechanical Engineering, University of Auckland, Auckland 1142, New Zealand; andrew.mcdaid@auckland.ac.nz; 3National Aeronautics and Space Administration, Ames Research Center, Moffett Field, CA 94043, USA; bill.toscano@nasa.gov

**Keywords:** human activity recognition, accelerometer, inertial measurement unit, wearable sensor, artificial intelligence, biomechanics, deep learning, convolutional neural network

## Abstract

Goal: To develop and validate a field-based data collection and assessment method for human activity recognition in the mountains with variations in terrain and fatigue using a single accelerometer and a deep learning model. Methods: The protocol generated an unsupervised labelled dataset of various long-term field-based activities including run, walk, stand, lay and obstacle climb. Activity was voluntary so transitions could not be determined a priori. Terrain variations included slope, crossing rivers, obstacles and surfaces including road, gravel, clay, mud, long grass and rough track. Fatigue levels were modulated between rested to physical exhaustion. The dataset was used to train a deep learning convolutional neural network (CNN) capable of being deployed on battery powered devices. The human activity recognition results were compared to a lab-based dataset with 1,098,204 samples and six features, uniform smooth surfaces, non-fatigued supervised participants and activity labelling defined by the protocol. Results: The trail run dataset had 3,829,759 samples with five features. The repetitive activities and single instance activities required hyper parameter tuning to reach an overall accuracy 0.978 with a minimum class precision for the one-off activity (climbing gate) of 0.802. Conclusion: The experimental results showed that the CNN deep learning model performed well with terrain and fatigue variations compared to the lab equivalents (accuracy 97.8% vs. 97.7% for trail vs. lab). Significance: To the authors knowledge this study demonstrated the first successful human activity recognition (HAR) in a mountain environment. A robust and repeatable protocol was developed to generate a validated trail running dataset when there were no observers present and activity types changed on a voluntary basis across variations in terrain surface and both cognitive and physical fatigue levels.

## 1. Introduction

Human activity recognition (HAR) has been well researched in the lab using both ambient and wearable sensors [1,2], including repetitive and single activities [3,4]. Chen [5] achieved a 93.8% accuracy using a smart phone for eight low intensity activities in a controlled environment, where the users self labelled activity with an app. Narayanan et al. reviewed 53 studies and found that with either one or two sensors, the accuracy ranged from 62% to 99.8% for posture and activity type in controlled environments [6]. It has been shown that gait is different when an athlete trains on stable verses unstable surfaces [7] and with variations in fatigue [8]. Ambient sensors include cameras and load cells which require the subject to perform in a fixed location, which is not possible in a mountain environment. Abnormal human activity recognition (abHAR) has been researched using video methods [3]. Inertial measurement units (IMU) are wearable sensors that can measure acceleration, rate of rotation and magnetic fields. IMUs have the advantage of being wireless, battery powered and small. Field based studies are more suited to accelerometers and IMU sensors where the subject may be out of sight. The number of sensors worn by a subject is a trade-off between the researcher requiring more detail and the subject complying while carrying out a task. In multiday scenarios, battery power and memory capacity start to constrain the number of sensors that can be used. Mountain trail running and military missions are long, fatiguing and sensitive to load, hence, they require light weight equipment, leading to a preference for a single accelerometer. Machine learning techniques have shown good accuracy when analyzing accelerometer data that are labelled, allowing supervised learning approaches. Narrow deep learning models with multiple layers and low numbers of neurons per layer, i.e., 200 vs. 2000, are optimal for battery powered applications due to the low memory usage and low computational requirements compared to deep learning models with more neurons per layer [1,9]. This paper covers the process of label allocation by the majority voting method and hyper parameter tuning of the narrow deep learning model, comparing various majority voting optimization, window size and overlap parameters. 

The question this paper investigated, is whether a deep learning model with only accelerometer data could perform at a sufficient accuracy given the data perturbation that occurred in trail running compared to lab-based data. In this study, we proposed a framework and workflow to calibrate a trail running track in the mountains to gather unsupervised data and assign verifiable labels to train a deep learning model. The data collection, labelling and model optimization are described in order to train a CNN and compare the results to training with a previously published lab-based dataset.

### 1.1. Related Studies

Various activities have been studied using accelerometers including: activity of daily living [10,11] with the UCI50 and Wireless Sensor Data Mining (WISDM) datasets [12]; factory workers [13]; food preparation [14]; tennis; snowboarding; weight lifting; rugby and running [15,16]. Johnson et al. [17] used cameras and load cells with deep learning to predict ground reaction forces. Xu et al. [18] modelled complex activities with 52 channels from seven IMUs. Buckley et al. [19] modelled a binary classifier for fatigue using three IMU sensors to classify fatigue or non-fatigue. The protocol measured athletes on two 400-meter runs on a track, separated by a beep test, until a self-reported Borg [20] scale exceeded 18. Buckley showed that IMU data can be used to determine biomechanical states over variations in fatigue. However, the labelling of data was trivial, given the protocol only determined when the two binary states took place during data collection. Bartlett et al. [21] used a single IMU on the thigh. However, the phase variable approach of the three defined lab-based activities was not applicable to fields with variable terrain. 

### 1.2. Labelling Data

Validation of activity, surface characteristics, and distance travelled in the lab occurs naturally, with the observation of the researcher to the prescribed activity and the surface of lab floor likely not being noted due to their homogeneity. Alternatively, users can label their own data when a smart phone app is used for labelling and accelerometry [5]. Activities in a remote field environment are self-selected, surface types change with location and speed is unpredictable, being determined by surface and fatigue levels. Furthermore, the user is occupied and cannot label data. This study outlined a protocol to address the challenges of non-observed field-based research to enable lab equivalent data validation and HAR detection accuracies.

### 1.3. Machine Learning

Deep learning automatically learns features in the data, replacing manually determined features and potentially using a larger number of features [13]. Many studies have investigated deep learning models through the utilization of previously published datasets based on the use of multiple sensors in a controlled environment. Wang [1] surveyed approaches in HAR, noting that deep learning techniques are replacing traditional pattern recognition techniques for repetitive activities such as running or walking, but are unable to determine activities such as having coffee. Nweke [4] reviewed deep learning algorithms for human activity using wearable sensors and found that they outperformed traditional machine learning models using manual features. He also concluded that variation in the data is required to make the model generalizable. Panwar [22] used a wrist sensor to measure reaching tasks on stroke patients, achieving an 88.9% accuracy on naturalistic data. Li [23] used a novel approach with a website, that continued to learn with the addition of an individual’s data, allowing variation in context with individual accuracy. However, this approach is not possible in a remote environment with no internet connection.

Various machine learning models have reported good results for HAR with deep learning models showing good results with their ability to automate the extraction of features. Wang surveyed the various models used for HAR, including convolutional neural networks (CNN), recurrent neural networks (RNN) (including the subset of long term short term memory LSTM) [24], deep belief networks (DBN), restricted Boltzmann machines (RBN), stacked autoencoder (SAE) and hybrid combinations of the above [1].

CNN has several advantages over other models, including local dependency, scale invariance and the capability to process multiple processors. Local dependency can take advantage of HAR signals which are related when close in time. Scale invariance means that patterns can be determined as they change over time, cadence or amplitude. RNN has also been used for HAR and its derivative long short term memory [24] with the limitation of not being able to use parallel processing and having stability problems.

Window sizes in the range of 2 to 10 s have been reported as giving the best results for HAR [10,25]. Wang et al. [10] discussed the trade-off between speed and recognition performance. Banos et al. [25] showed that window sizes of less than 2 s gave good a detection performance. There are varying requirements for single instance activities (e.g., climbing a fence) compared to repetitive activities (e.g., running, walking). Overlap can allow for an increased classification accuracy of time series data. A strategy must be chosen to determine the label for a given window of multiple samples when each sample is individually labelled. 

The number of samples used in this work is greater—by a factor of three—than in the WISDM dataset [26] (Wireless Sensor Data Mining). Datasets in the lab often have multiple subjects to address generalizability and statistical significance. We propose that uneven terrain, real world obstacles and fatigue are important contexts in human recognition to determine the accuracy of models out of the lab. This work will present a single participant to determine the validity of the approach replacing intersubjective variation with terrain, obstacles, and fatigue. 

This study presents a dataset and HAR model in a mountainous environment on a fatiguing subject, where transitions in activity are determined by the subject’s spatial location and self-directed choice rather than time.

## 2. Materials and Methods

This paper provides a novel dataset and experimental protocol. A mountain trail was calibrated with waypoints and labels for any transition in terrain texture (concrete, gravel, grass, mud, rocks, rivers) and slope. The participant activity was self-selected by the subject in the mountains, unobserved, and the subject increasingly became fatigued (both physically and mentally) throughout the study. Single instance activities (e.g., climb gate) are included with repetitive activities (e.g., lay, sit, walk, run). An overview of the method is described in Figure 1, with Table 1 describing the steps in pseudo code.

### 2.1. Ethics

The researcher’s university ethics committee (AUTEC 18/412) approved all procedures in the study and the participant provided written informed consent prior to participating in the study.

### 2.2. Mountain Trail Characteristics and Calibration

A 3.8 km mountain trail with a total of 194 m vertical elevation and duration of 25 to 35 min for a fit healthy adult was selected. The route was chosen such that it had various surfaces and slopes, with the same start and end location. 

The trail was divided into segments defined by waypoints (Table 1 point #1) where each segment had a feature change, such as terrain, obstacle or slope (Figure 2a). Reference information included videos (GoPro Hero 4, Garmin Kansas, Olathe, KS, USA) for terrain and obstacles with a topographic map (source: www.linz.co.nz) for calculating the slope. The course was segmented, with a GPS location and altitude designating the start of each segment (Table 1 point #2). A segment was started if a feature changed, including slope (up, flat, down), surface texture (concrete, gravel, mud, grass) or obstacles (river, gate, stairs). Small obstacles such as rocks, branches or holes in the track did not require a new segment. Each waypoint was located on Google Earth satellite view and topographic map data (source: www.linz.co.nz) using vegetation or ground features for registration and manually recorded to six decimal places, equating to a 10 cm accuracy (Figure 2b). A comma separated variable file was generated with columns for waypoint number, waypoint name, latitude, longitude, altitude, terrain slope in degrees, and terrain texture. The initial labeling of the course and activity labels was performed by the participant and validation was performed by two additional researchers using videos (GoPro Hero 4, Garmin Kansas, USA) for the terrain type and obstacles, with the addition of topographic maps for slope calculations. Validation of activity was performed with a summary of the plots shown in Figure 3 The correctness of the labeling was an interesting aspect of this research and was determined by comparison of three researchers’ opinions with a particular focus on transitions, such as approaching an obstacle, as shown in Figure 3.

### 2.3. Sensors on the Subject

The subject wore a BioHarness [27,28] around the chest (Medtronic, Dublin, Ireland for acceleration data (sample rate 100 Hz, x-axis = vertical, z-axis = sagittal and y-axis = lateral) and a Garmin Forerunner GPS watch (Garmin, Olathe, KS, USA) on the wrist (sample rate 1 Hz, horizontal accuracy 6 m) to assist with the labelling of location.

### 2.4. Physical and Cognitive Load Protcol

The protocol was designed to induce fatigue over multiple hours and was broken into hourly segments which could be viewed as three sections. Firstly, physical load, followed by cognitive load and finally, a small rest period before restarting at the top of the hour. The physical load performed was trail running with obstacles (25 to 35 min depending on level of fatigue), immediately followed by a 15 min cognitive fatigue test battery (Figure 4).

### 2.5. Assessment Battery Tests for Cognitive Load

A custom app (labelled ‘COGNI’) was developed to be performed on an iPad Pro (Apple, Cupertino, CA, USA). The test battery covered the expected types of cognitive fatigue by including; Stroop [29,30] for cognitive control, Finger Tap Test [31] for neuromuscular control, Trail Making A [32,33] for response timing for sequence tracking, Trail Making B for response timing for sequence tracking with divided attention and Spatial Memory for working memory. A custom implementation of the Multi Attribute Test Battery (MATB) [34,35] was performed on an Apple MacBook Pro (Apple, Cupertino, CA, USA) running in Python. The COGNI testing was completed twice per hour with the MATB assessment in between (Figure 4).

For determining ground speed, the participant was instructed to “go as fast as possible” and repeat the trail run 24 times over a 24-h period. This protocol was part of a larger study on the effects of fatigue on performance. The results of the various cognitive tests are not reported in this paper. The participant could not continue due to physical exhaustion after 11 h, demonstrating that the protocol induced extreme levels of fatigue, i.e., from no fatigue through to complete fatigue. 

### 2.6. Dataset Preparation for HAR

The aim of the data processing was to generate a dataset with columns for the subject, timestamp, acceleration vertical, acceleration sagittal, acceleration lateral and activity label. Subject was the person’s anonymized number, the timestamp was the date and time to an accuracy of milliseconds, the acceleration channels were obtained directly from BioHarness and the label was the activity label with a resolution of 1 s.

Location of the subject on the track was determined by analyzing the GPS, Table 1 point #4, to derive the time of closest proximity to a waypoint. The time at each waypoint was determined when the subject to waypoint proximity changed from approaching to leaving.

Activity labeling started with feature extraction of cadence in steps per minute based on a zero crossing of the vertical acceleration (100 < walk < 150, run > 150). Acceleration data were normalized (as shown in Table 1 point #5), where the same minimum and maximum for all three axes was applied to maintain scale relationships (point #6). Activity type was further refined based on knowledge of the protocol sequence, the track, waypoint proximity times (point #7), and observation of the acceleration waveforms. Figure 3 shows an example of running up to a gate, with a transition to walking before climbing the gate, followed by walking then running. 

Feature columns were combined to derive multimodal features in a single column, such as terrain derived slope and activity, to enable “run-up” vs. “walk down” (Table 1 point #8). The 100 Hz acceleration data were allocated an activity label per row by time, synchronizing segments of samples between each waypoint using the asynchronous labelled location data.

Data were reduced from 3,829,759 samples to 3,341,184 samples by removing data that did not have the following labels (lay, sit, climb gate, walk, run) and during rest periods where activity was not prescribed or recorded (Table 1 point #9). A comparison with similar activity labels and datapoints is shown with the WISDM dataset in Table 2. The comparison WISDM dataset by Kwapisz et al. 2010 [26], has 1,098,207 samples over six attributes, with 33 participants. The trail run variations were due to terrain and fatigue levels, whereas the WISDM dataset variations were derived from multiple subjects for homogenous surface and fatigue.

### 2.7. CNN Data Preparation

CNN models require segments of data to be input during training. Time series data with n samples were divided into windows of W samples wide with S overlap leading to D rows of data (Equation (1)).
(1)$$D=1+\;\frac{\left(n\;-\;W\right)}{S}$$

Each data point in the window can have different labels, however a decision is made for one label per window using a Majority Voting method with a THreshold (MVTH), treated as a hyper parameter in the training model. Majority voting can assist when activities change and a dataset includes two activities, whereas the CNN classifier is required to choose one class. Labelled windows were excluded if the majority label vote did not exceed a threshold. For example, MVTH = 0.4 majority voting would exclude a window if 40% of the samples where not identical. 

Time series data were transformed for CNN model compatibility into an array (D,W,F) with D rows, W samples wide with number of features (F) equal to 3 from the accelerometer axis (x,y,z). A randomized train test split of 0.33 was selected using sklearn in python.

Tuning of the window size and window labelling rejection ratio was performed to optimize accuracy between repetitive activities (e.g., walking) and one-off activities (e.g., climbing gate).

### 2.8. Deep Learning Model

For classification, a CNN topology was used, as shown in Figure 5. This network consisted of three separate 1D convolutional networks for each acceleration axis, joined by a dense layer to achieve a multivariate classification. Each axis included two convolutional 1D layers with filter size 64, kernel size 3 and ReLu activation. This was followed by a drop out layer set at 50% for generalizability and a max pool layer with pool size 2. The three separate channels were combined in a flattened layer, followed by a dense layer using a ReLu activation and finally, a softmax activation function to give a probability density function for each class. Learning used an epoch of 100 with batch size of 50 and an Adam optimizer (lr = 0.001, beta_1 = 0.9, beta_2 = 0.999, epsilon = 1 × 10^−8^). Data were split: 0.67 for training and 0.33 for testing. The classification accuracy was calculated for each label as a multivariate analysis.

## 3. Results

### 3.1. Accelerometer Data over Various Surfaces

Various running surfaces with changes in slope, texture and obstacles were expected to give increased gait variability. Figure 6 illustrates how the accelerometer waveforms changed for a subject running up and down a hill (slope between 8° and 10°) and how these changed when travelling over a hard road surface versus a soft uneven track surface. The waveforms in Figure 6 illustrate the difference in gate between running up and down, with a large amount of deceleration evident when landing on a down hill slope. The waveforms in Figure 6 show how terrain modulates the acceleration of the participant with more variation and lower amplitudes, possibly due to softer and less uniform surfaces. Zero crossing statistics are summarized in Table 3 to highlight the variation in timing across different terrains and slopes, showing the difference between hard smooth terrain compared to soft rough terrain. When running down, the standard deviation increases by a ratio of 19.4. These waveforms indicate the complexity of the algorithmic task to determine activity classification.

### 3.2. Experiment One—Majority Voting Optimization

(1) Methods

The majority voting threshold, MVTH, for each window was tested from a value of 0.20 to 0.95 for labels (lay, sit climb gate, walk, run), epoch 100, batch 50, window size 256, overlap 128, windows 26201. 

(2) Results

A minimum accuracy of 0.973 was achieved with MVTH 0.20 and 0 rejected windows. The accuracy increased monotonically and was flat above MVTH 0.8, accuracy 0.982, rejected windows 494. As such, we chose 0.2 and 0.8 in experiment two.

### 3.3. Experiment Two—Window Size and Overlap Optimisation

(1) Methods

A second experiment was run with MVTH 0.2 and 0.8, with a range of window sizes (W) and overlap (S), as shown in Table 4. Note, window size determines the number of neurons used in the models input layer.

(2) Results

The highest overall accuracy 0.982 was for W 256, S 128 and MVTH 0.8; however, the “climb gate” precision was highest for W 128, S 64 and MVTH 0.80 with the accuracy dropping by 0.04 to 0.978. The tradeoffs can be seen in Figure 7.

Figure 7 shows the trade off for window size, where the one off activity “climb gate” is optimized at 128 samples, while the periodic activities are less sensitive to windows size. 

## 4. Discussion

A protocol is presented to calibrate an outside trail running track with novel data segmentation and labelling methods. To our knowledge, this is the first time a dataset has been collected in an unstructured mountainous environment and validated with a deep learning model. This work will permit the completion of further research in applications outside of the lab environment, in rough terrain and with voluntary movement over time.

A CNN deep learning model was used as a multi variate HAR classifier on a dataset that was obtained from a trail run. The protocol included voluntary transitions between activities, various obstacles, variations in surface texture, terrain slope and subject cognitive and physical fatigue. The three axis 100 Hz acceleration data were labelled by activity with a temporal resolution of one second. The resulting time series was sliced into windows, where the majority voting method assigned a single label per window. The resulting array was passed to a CNN multivariate classifier and trained to learn activity from the acceleration signals. Increasing the value of the majority voting threshold improved the accuracy for all five classes of activity with no improvement for threshold values over 0.8. The dataset included repetitive activities and single instance activities which required hyper parameter tuning in order to reach an overall accuracy of 0.978, with a minimum class precision of 0.802 for the one-off activity. 

The minimum precision of the laboratory experiment from the WISDM dataset was 0.928. Results [36] using deep bidirectional long term short term memory combined with CNN resulted in an accuracy of 0.980, which may not be feasible for battery powered field devices. Additionally, using a two channel CNN resulted in an accuracy of 0.953, with both channels using the UCI public dataset on human activity recognition [37]. Wang [1] reviewed 53 HAR studies, finding that 80% of studies exceeded an 85% accuracy level. Quantifying what is acceptable, the comparative accuracy for this paper was set at 0.95. Hence, the trail running CNN model accuracy showed that deep learning HAR could be performed accurately in the field with an appropriate protocol to enable the labelling of repetitive activities. There was a trade off in window size for repetitive activities versus one off activities, with the result being a window size of 128 to achieve a “climb gate” precision level of over 80%, with the overall accuracy only reducing by 0.4%.

From these results, it can be confirmed that variations in terrain with obstacles and cognitive and physical fatigue can be incorporated into a protocol outside of the laboratory setting with self-selected activity when using a single sensor and a deep learning model.

### Limitations

This work was limited to one participant. For further work, it is recommended that the models be extended to include multiple participants, using the protocol described. Deep learning models can be personalized and this step should represent the next stage for this model [38]. Comparison with other deep learning models was not within the scope of this study. A comparative analysis used the term “acceptable accuracy” based on recently published results and systematic reviews. Additionally, sleep deprivation should be induced by reducing the hourly physical load, which in turn would allow for a longer period in which to complete the protocol. GPS accuracy in tree covered deep valleys was reduced and installing a video camera on each person for every run would speed up the manual labelling process. Furthermore, computer vision could be used to recognize waypoints.

The effectiveness of the protocol was encouraging, as it enabled translational research to be undertaken in environments that are closely related to where they would have the most impact. For example, soldiers’ workload and training on multiday missions, can determine activity types allowing for adjustments to be made to team workload, maximizing speed and reducing the risk of injury. Adventure sports enthusiasts can gain insights into their pace and activity types in order to improve training and reduce the risk of injury. This protocol enables performance analyses for the likes of selection in military applications or activity recognition for remote workers in dangerous environments. Additionally, ambient sensing is a key consideration for health and performance decision making for deep space crews.

## 5. Conclusions

This paper presents a framework which includes a track calibration protocol, data collection protocol, data analysis pipeline, mixed label optimization method and CNN model tuning procedure. The framework may allow the calibration of a trail running track in the mountains in order to facilitate an activity protocol with labelled data, where each activity type is self-selected over time. Further work in different locations and across more participants is required. This dataset has similar classes of activity to lab data, but includes an intentionally induced modulation in gait, using the environment and level of fatigue as stimuli.

A multi variate CNN deep learning model was implemented on time series data with hyper parameter tuning to maximize the overall accuracy (0.978) and individual class precision (0.801–0.988). 

In the experiments conducted during the study, we confirmed that a deep learning model could accurately classify activities from an accelerometry dataset from a trail run with modulated terrain, slope and subject fatigue level. The results showed similar accuracy and precision results to equivalent datasets in a controlled laboratory environment.

The primary contributions of this work are, firstly, the capability of calibrating an outside track for field-based experiments, and secondly, the development of a HAR model for mountain tracks with surface texture and fatigue as the modulating effects.

In future work, we will conduct further studies over multiple days to include sleep deprivation in the fatigue protocol and increase the number of subjects to assess the generalizability of the model. Further work could also assess if ensemble models enable variable window sizes to attain higher precision levels for one off activities. 

## Figures and Tables

**Figure 1 sensors-21-00654-f001:**
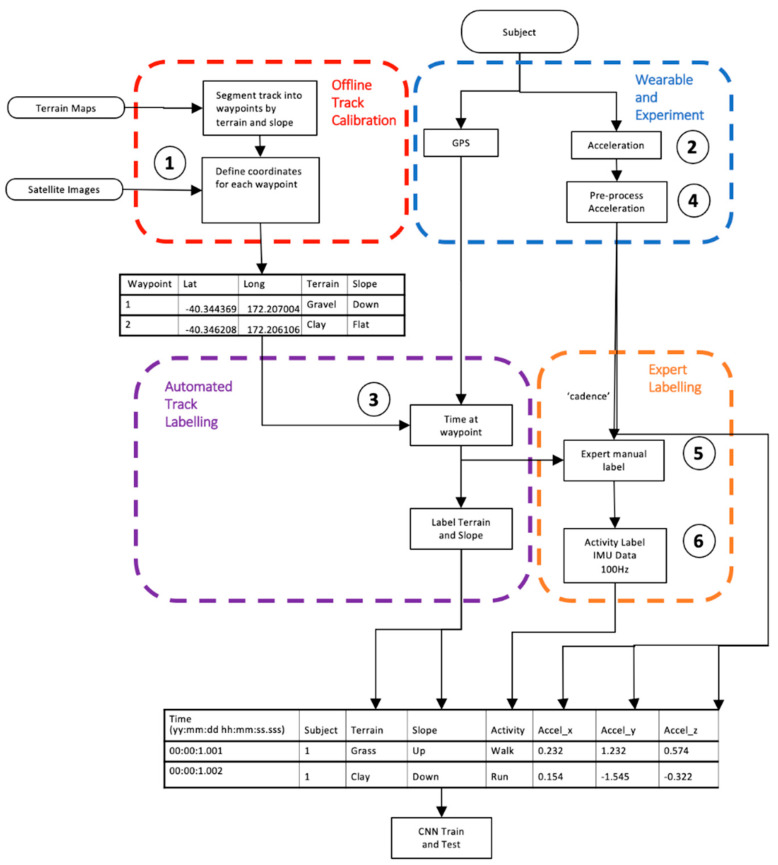
Data pipeline for trail calibration and sensor fusion.

**Figure 2 sensors-21-00654-f002:**
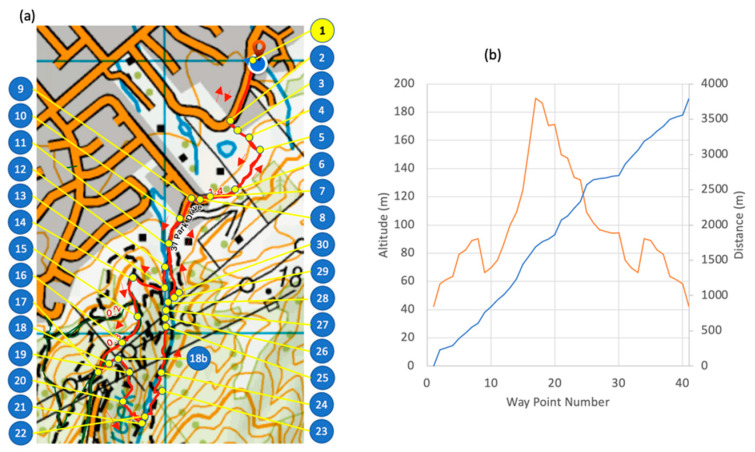
(**a**) Track calibration with segmentation waypoints and (**b**) ground features validation.

**Figure 3 sensors-21-00654-f003:**
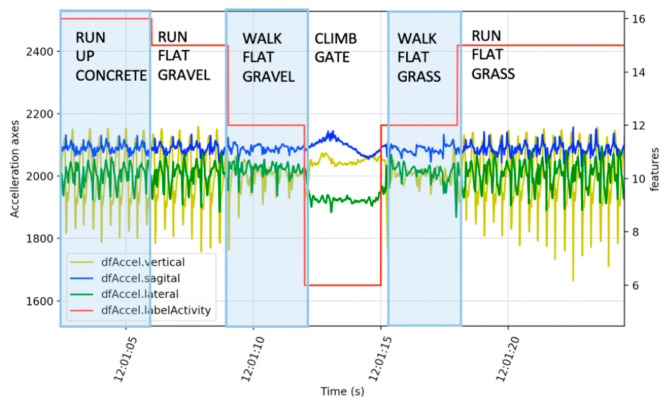
Time series example of the one-off activity “climb gate” verses repetitive data “run” and “walk”.

**Figure 4 sensors-21-00654-f004:**
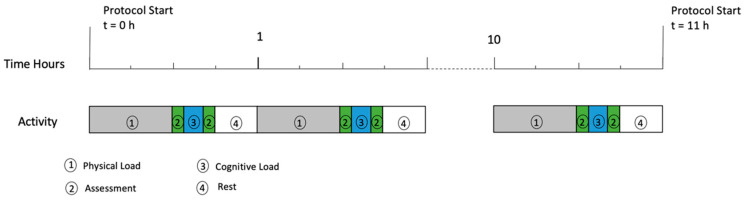
Protocol description for the physical and cognitive load during the experiment.

**Figure 5 sensors-21-00654-f005:**
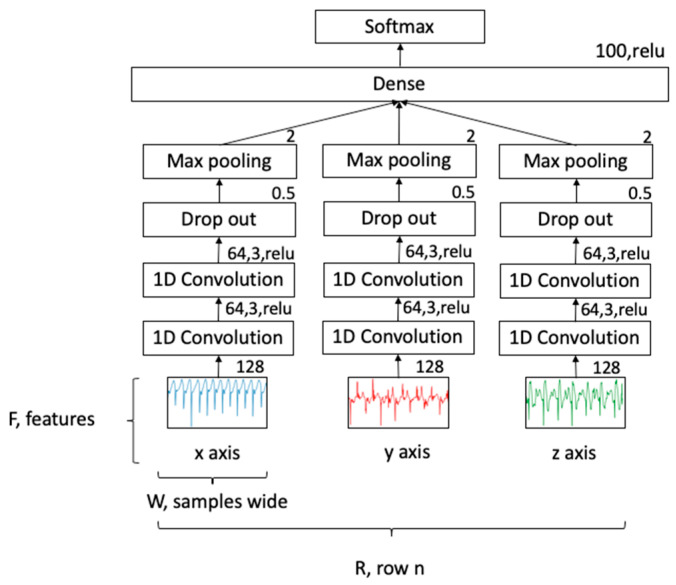
CNN structure.

**Figure 6 sensors-21-00654-f006:**
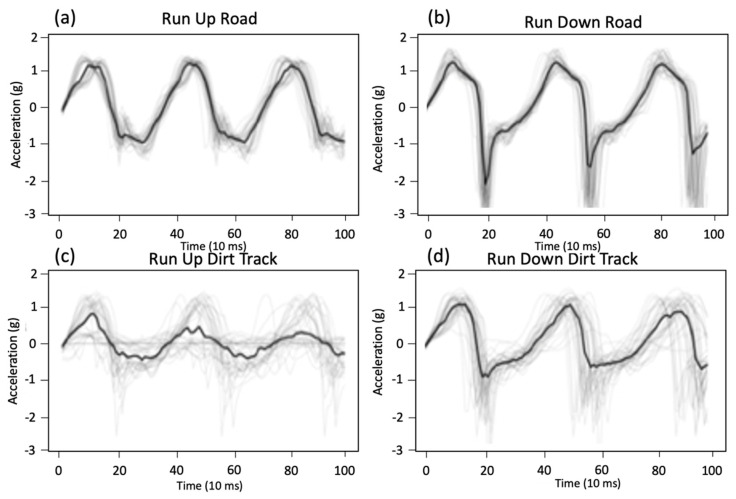
Acceleration waveforms for running up and down slopes over various terrain (road, track), (**a**) run up road, (**b**) run down road, (**c**) run up dirt track, (**d**) run down dirt track.

**Figure 7 sensors-21-00654-f007:**
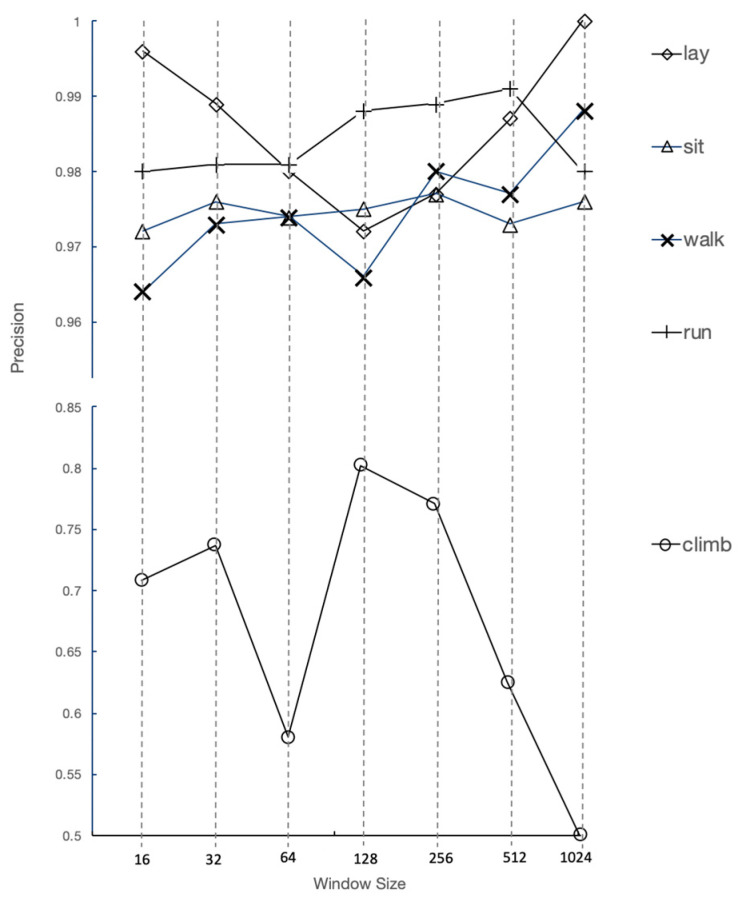
Precision by window size results for trail run activities (lay, sit, walk, run, climb) from the for CNN with MVTH = 0.8.

**Table 1 sensors-21-00654-t001:** Pseudo code of the data pipeline for trail calibration, protocol, data analysis.

	Process−
1	Define a course by waypoints, *WP*(n,x,y,z) where x,y,z are (latitude, longitude and altitude) and n = 1,2..N. Each waypoint defines a change in Terrain (concrete, grass, clay), Slope (down steep, down, flat, up, up steep) and Object (fence, gate, river, stairs). Define the same start and end waypoint, *WP*(1) = *WP*(N).
2	Define the location along the course, *GPS*(t,x,y,z), where t_s_ = 1,2,... seconds and x,y,z are latitude, longitude and altitude. Define the 3 axis accelerometer data along the course, *ACC*(x,y,z,t_ms_), where t_ms_ = 1,2,.. ms GPS(t,x,y,z) is the location over time of the participant, each lap starts at T_lap_start_ and finishes at T_lap_stop_, so for a lap, t = T_lapstart_,…,T_lap_stop_. Align time stamps between sensors by calculating time offset. Manual correction is performed by an expert observing the acceleration waveforms.
3	Collect GPS and accelerometer data from participant during experiment.
4	Define the time at each waypoint *T*(N) = min( *WP*(N,x,y,z) − *GPS*(t,x,y,z), t = T_lapstart_,…,T_lap_stop_)
5	Normalize date to maintain inter axis scale relationships ACCmin = min (ACCx,ACCy,ACCz) ACCmax = max(ACCx,ACCy,IMYz) IACCnormal (x,y,z) = (ACC(x,y,z) − ACCmin) / (ACCmax − ACCmin)
6	Pre-process including calculating zero_crossings = (ACC(y,t_ms_) = 0) *CADENCE*(t) = 60/( zero_crossings(n) − zero_crossings(n − 1) )
7	Observe *ACC*(x,y,z,t_ms_) and *CADENCE*(t) to label activity at each time step. Allocate ACTIVITY(t) based on observation.
8	Allocate labels to the accelerometry array, ACC, for terrain, slope and activity type, ACC (activity, terrain, slope, x, y, z, t_ms_)
9	Remove data not allocated to required labels
10	Train the CNN model using ACC(x,y,z,t) against ACTIVITY(t)

**Table 2 sensors-21-00654-t002:** Dataset samples count by label.

Label	Trail Running	Label	WISDM
Climb Gate	26,099	Sit	59,939
Lay	70,100	Stand	48,395
Sit	1,065,100	Down Stairs	100,427
Walk	741,599	Up Stairs	122,869
Run	1,438,302	Walk	424,398
		Jogging	342,176
Total	3,341,184	Total	1,098,204

**Table 3 sensors-21-00654-t003:** Vertical axis zero crossing time for two activities and two terrain surfaces.

Activity	Terrain Surface	
	Road	Track
	(s)	(s)
Run Up	0.365 ± 0.013	0.381 ± 0.030
Run Down	0.352 ± 0.014	0.475 ± 0.272

**Table 4 sensors-21-00654-t004:** Results for the trail run for CNN over stride and label rejection ratio for activities (lay, sit, walk, run, climb). Grey highlight indicates rows for maximum accuracy by activity and best overall accuracy.

Window Size W	Overlap S	Ratio MV_TH_	Windows Accept D	Windows Reject D	Accuracy	Precision				
						Lay	Sit	Climb Gate	Walk	Run
16	8	0.2	417647	0	0.972	0.996	0.970	0.674	0.964	0.979
16	8	0.8	417043	604	0.973	0.996	0.972	0.708	0.964	0.980
32	16	0.2	208823	0	0.973	0.991	0.968	0.702	0.972	0.981
32	16	0.8	208313	510	0.977	0.989	0.976	0.737	0.973	0.981
64	32	0.2	104411	0	0.973	0.972	0.975	0.664	0.967	0.980
64	32	0.8	103979	432	0.974	0.980	0.974	0.580	0.974	0.981
128	64	0.2	52205	0	0.973	0.988	0.978	0.702	0.954	0.982
128	64	0.8	51721	484	0.978	0.972	0.975	0.802	0.966	0.988
256	128	0.2	26102	0	0.973	0.929	0.984	0.700	0.956	0.981
256	128	0.8	25608	494	0.982	0.977	0.977	0.771	0.980	0.989
512	256	0.2	13050	0	0.970	0.921	0.990	0.773	0.939	0.976
512	256	0.8	12613	437	0.981	0.987	0.973	0.625	0.977	0.991
1024	512	0.2	6524	0	0.953	0.971	0.964	0.455	0.884	0.986
1024	512	0.8	6135	389	0.980	1.000	0.976	0.500	0.988	0.980

## Data Availability

WISDM dataset can be downloaded from https://www.cis.fordham.edu/wisdm/dataset.php.

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
