# Peer review of "Moving the Lab into the Mountains: A Pilot Study of Human Activity Recognition in Unstructured Environments"

_sensors, 2021, doi:10.3390/s21020654_

Round 1

Reviewer 1 Report

The authors present a protocol for constructing human activity recognition data in the wild, with a single participant logging triaxial accelerometer and GPS data while trail running in the mountain. The trail running dataset was evaluated using various CNN models constructed by varying accelerometer data’s window sizes and overlaps. The proposed protocol for gathering data in the wild seems to be effective, and I believe many will be interested in replicating or mimicking the protocol to suit their data gathering needs. However, in order to replicate the data gathering protocol, the details given in the manuscript must be clear and accurate. Please review the following issues and revise as necessary.

  • It is unclear whether the human data logger (who conducted the trail running) participated in the labeling process. Did the third person infer the activity labels based on the accelerometer, terrain, and slope information? This information of who actually labeled the data is important and should be clearly stated. Moreover, readers would also want to know how the correctness of the imparted activity labels was verified.
  • Section 2 is the core of this manuscript. Please re-examine Table 1.
    • Point #4. The WP(n,x,y,z) and GPS(t,x,y,z) are both location data. I don’t understand how subtracting and taking the minimum (of x, y, z values?) can bring about the time at each waypoint. The main text (lines 187-189) gives an explanation, but the equation should be revised since it is confusing.
    • Point #5. IMUmin equation contains an extra IMU sign.
    • Point #6. Cadence refers to number of steps per minute. Based on the definition, I read the equation to be counting the number of zero-crossings occurring at the y-axis per second. (There are 100 data points per second.) The description given in the main text, however, does not provide clear explanation of #6 (see lines 192-193).
    • Points #7 and #9. Activity labels are allocated at two steps. Why?
    • Point #10. Lines 203-204 does not match #10. (Seems to match #9.)
  • Line 234. Max pool layer with pool size 0.5 seems to be incorrect. It also does align with the Figure 5. (‘2’ is written above the max pooling box.)
  • Line 258. The expression ‘stride 256’ should be corrected to ‘window size 256’.
  • Table 4 should be re-examined. The ‘Window Accept’ and ‘Window Reject’ header should be better written as ‘Data Accept’ and ‘Data Reject’. The two headers should have ‘D’ as a second header, which aligns with equation 1 in page 8. Calculating the number of data based on eq. 1 in page 8 gives different numbers than Table 4. Please double check.
    • 417648 → 417646 / 208823 → 208822 / 104411 → 104410 / 52205 → 52204 / 26102 → 26101
  • The following are a list of typos that were spotted:
    • Line 85: out performed → outperformed
    • Line 175: Missing double quotes. “go as …
    • Line 201: in a single ‘column’ → in a single ‘row’ (I think)
    • Line 202: Missing single quote after ‘walk down
    • Line 219: (THMV) → (MVTH)
    • Line 249: software → soft or softer
    • Line 289: dataset min precision → dataset’s minimum precision
    • Figure 2. Inside the figure: ‘Altimeter is un accurate.’ Please revise the captions inside the figure.
  • The use of the term ‘IMU’ in the paper is incorrect. Inertial measurement units (IMU) are composed of accelerometers, gyroscopes, and magnetometers which measure linear acceleration, angular velocity, and magnetic field strength (to orient with respect to the Earth’s axes), respectively. (https://www.sciencedirect.com/topics/engineering/inertial-measurement) The BioHarness logs triaxial accelerometer data only.
  • In subsection 2.3, the authors cite BioHarness as the device for logging activity information. I’ve checked BioHarness’s two references [26, 27], and found that the device contains “Tri axial ACC (accelerometry), using piezoelectric technology (i.e. cantilever beam set up) which samples at 18 Hz”. However, the authors state the sample rate to be 100 Hz in the manuscript (line 153). Please explain the discrepancy.

Reviewer 2 Report

The authors proposed human activity recognition via deep learning with single sensors in the mountains, such as field-based activities. The article is well-explained and clearly demonstrated the experimental analysis.

However, several comments are listed as follows:

  • For the dataset, the training-testing protocols are not well-defined. For instance, how many samples are used for training, validation and testing? This could help the researchers who are interested in using the dataset as a guide.
  • What is the feature input size for the CNN? is it 1×128 for each feature? It showed in Figure 5, but not in the main text.
  • During the training configurations, what is the learning rate? What kind of optimizer is implemented? What is the weight decay?
  • The authors mentioned that they combined/joined the flatten layers into a dense layer -> do you mean concatenation operator? Is this Siamese network?
  • The authors have sufficient proof for the experimental results; however, the reviewer would like to see the comparisons with other approaches, such as ResNet, multi-stream CNN, Siamese Network, etc. This would be interesting in your conclusion.
  • Minor changes:
    • Please rescale Figure 3b.
    • Figure 6 is blurry. Please modify in the revised version

Reviewer 3 Report

The authors intend to create and validate a field-based data collection and assessment method for human activity recognition in the mountains with variations in fatigue with a single sensor and a machine learning model. For the acceptance, the paper may be reworked and well-written. In general, the paper has several flaws that are specified as follows:

  1. Abstract should be reworked for better understanding. I agree that you intended to do a structured abstract, but it is difficult to read.
  2. The language of the paper should be refined;
  3. The related works section must be reworked for better reading with tables or images;
  4. As well as, there are other studies that may be included more recent;
  5. The remaining subsections of the introduction may be embedded in other sections, because they are not the core of the paper;
  6. Section 2 is very well-structured;
  7. Section 3 should be improved and detailed with more explanation about the results, images, diagrams, tables and so on.
  8. Section 4 may be included in the results as "Results and Discussion";
  9. Conclusions are well-written;
  10. The author contributions may be improved.

Round 2

Reviewer 2 Report

After reading the response to the initial report, the reviewer still has concerned about the performance comparisons between the proposed CNN model with other approaches.

Without additional experimental analysis, it is difficult to conclude that the accelerometer and proposed CNN model could achieve acceptable performance in unconstraint environments. 

Reviewer 3 Report

Congratulations, the authors correctly improved the manuscript.

Author Response

thank you, for all your feedback